# Protective Effect of Low-Dose Alcohol Consumption against Post-Ischemic Neuronal Apoptosis: Role of L-PGDS

**DOI:** 10.3390/ijms23010133

**Published:** 2021-12-23

**Authors:** Chun Li, Jiyu Li, Ethyn G. Loreno, Sumitra Miriyala, Manikandan Panchatcharam, Hong Sun

**Affiliations:** Department of Cellular Biology & Anatomy, Louisiana State University Health Sciences Center-Shreveport, Shreveport, LA 71103, USA; chun.li@lsuhs.edu (C.L.); jiyu.li@lsuhs.edu (J.L.); elo001@lsuhs.edu (E.G.L.); sumitra.miriyala@lsuhs.edu (S.M.); manikandan.panchatcharam@lsuhs.edu (M.P.)

**Keywords:** ethanol, brain, ischemic stroke, apoptosis, L-PGDS

## Abstract

Ischemic stroke is one of the leading causes of permanent disability and death in adults worldwide. Apoptosis is a major element contributing to post-ischemic neuronal death. We previously found that low-dose alcohol consumption (LAC) protects against neuronal apoptosis in the peri-infarct cortex following transient focal cerebral ischemia. Lipocalin-type prostaglandin D2 synthase (L-PGDS), which is mainly localized in the central nervous system (CNS), was previously shown to inhibit neuronal apoptosis. Therefore, we determined whether L-PGDS is involved in the protective effect of LAC against post-ischemic neuronal apoptosis. Wild-type (WT), CaMKIIα^CreERT2/+^/L-PGDS^+/+^, and CaMKIIα^CreERT2/+^/L-PGDS^flox/flox^ mice on a C57BL/6J background were gavage fed with ethanol or volume-matched water once a day for 8 weeks. Tamoxifen (2 mg/day) was given intraperitoneally to CaMKIIα^CreERT2/+^/L-PGDS^+/+^ and CaMKIIα^CreERT2/+^/L-PGDS^flox/flox^ mice for 5 days during the fourth week. AT-56 (30 mg/kg/day), a selective inhibitor of L-PGDS, was given orally to AT-56-treated WT mice from the fifth week for four weeks. Cerebral ischemia/reperfusion (I/R) injury, TUNEL-positive neurons, and cleaved caspase-3-positive neurons were measured at 24 h of reperfusion after a 90 min unilateral middle cerebral artery occlusion (MCAO). We found that 0.7 g/kg/day but not 2.8 g/kg/day ethanol significantly upregulated L-PGDS in the cerebral cortex. In addition, 0.7 g/kg/day ethanol diminished cerebral ischemia/reperfusion (I/R) injury and TUNEL-positive and cleaved caspase-3-positive neurons in the peri-infarct cortex in WT and CaMKIIα^CreERT2/+^/L-PGDS^+/+^ mice. Furthermore, the neuroprotective effect of 0.7 g/kg/day ethanol was alleviated in AT-56-treated WT and CaMKIIα^CreERT2/+^/L-PGDS^flox/flox^ mice. Our findings suggest that LAC may protect against cerebral I/R injury by suppressing post-ischemic neuronal apoptosis via an upregulated L-PGDS.

## 1. Introduction

Ischemic stroke, which occurs when an artery supplying blood to the brain becomes obstructed, is one of the major causes resulting in permanent disability and death globally [1,2]. Due to the use of tissue-type plasminogen activator (tPA) and intravascular techniques, transient focal cerebral ischemia has become a common type of ischemic stroke. However, it paradoxically leads to cerebral ischemia/reperfusion (I/R) injury [3]. Apoptosis, a form of programmed cell death, contributes to a significant proportion of neuron death in the ischemic penumbra during reperfusion [4]. Apoptosis is indicated by a series of typical morphological features, including the formation of apoptotic bodies, chromatin condensation, and cell and nuclear shrinkage. A biochemical hallmark of apoptosis is the formation of DNA fragments. To date, three apoptotic pathways (the perforin/granzyme, extrinsic (death ligand), and intrinsic (mitochondrial)) have been discovered. Caspase-3 appears the common effector caspase in all three apoptotic pathways [5]. It is activated upon proteolytic cleavage via the intrinsic and extrinsic pathways following ischemic stroke [6].

Alcohol is one of the most commonly and regularly used chemical substances among humans. Epidemiological studies have found that mild-moderate ethanol consumption may reduce the morbidity of ischemic stroke and decrease infarct volume and mortality from ischemic stroke [7,8,9,10,11,12]. Consistently, we recently found that low-dose alcohol consumption (LAC) significantly reduces cerebral I/R injury in rodents [13,14]. Moreover, the reduced cerebral I/R injury appears to result from suppressed inflammation and neuronal apoptosis [14,15]. L-PGDS is known as the principal PGH_2_ isomerase and transporter of small lipophilic molecules, expressed mainly in the central nervous system (CNS) and cardiovascular system [16,17]. L-PGDS catalyzes the conversion of PGH_2_ to PGD_2_, which is the most abundant prostaglandin produced in the brain [18]. A previous study reported that genetic knockout of L-PGDS significantly increased infarct volume following either transient or permanent focal cerebral ischemia [19]. Recently, L-PGDS was found to be associated with vagus nerve stimulation-induced suppression of post-ischemic neuronal apoptosis [20]. Chronic alcohol consumption was reported to upregulate L-PGDS mRNA in adipose tissue [21]. Thus, our first goal was to determine whether chronic alcohol consumption alters L-PGDS in the brain. Since we found that LAC but not heavy alcohol consumption (HAC), which exacerbates cerebral I/R injury, alters L-PGDS in the brain, our second goal was to determine whether L-PGDS is involved in LAC-induced suppression of neuronal apoptosis following transient focal cerebral ischemia.

## 2. Results

### 2.1. Control Conditions

As shown in Table 1, LAC (eight weeks of daily feeding with 0.7 g/kg/day ethanol), HAC (eight weeks of daily feeding with 2.8 g/kg/day ethanol), four weeks of daily treatment with 30 mg/kg/day AT-56, and five days of daily treatment with 2 mg/day tamoxifen did not significantly change physiological parameters, including body weight, mean arterial blood pressure (MABP), and heart rate.

### 2.2. Effect of Alcohol on L-PGDS

As shown in Figure 1, 8-week LAC significantly upregulated L-PGDS by 1.4-fold in the cerebral cortex under basal conditions. In contrast, 8-week HAC did not significantly alter protein expression of L-PGDS in the cerebral cortex. 

### 2.3. Effect of Chronic Alcohol Consumption on Cerebral I/R Injury

As shown in Figure 2, eight-week LAC before MCAO significantly reduced the infarct volume and improved somatic sensorimotor function at 24 h of reperfusion in WT mice. Four-week administration of AT-56 did not further exacerbate 90 min MCAO/24 h reperfusion-induced cerebral damage in water-fed WT mice but significantly alleviated the neuroprotective effect of LAC (Figure 2A,B). As shown in Figure 3A,B, the protein expression of L-PGDS in the cerebral cortex was remarkedly reduced in tamoxifen-treated CaMKIIα^CreERT2/+^/L-PGDS^flox/flox^ mice compared to tamoxifen-treated CaMKIIα^CreERT2/+^/L-PGDS^+/+^ mice, indicating the success of neuron-specific L-PGDS conditional knockdown in the forebrain. The neuroprotective effect of LAC was also found in CaMKIIα^CreERT2/+^/L-PGDS^+/+^ mice (Figure 3C–E). Forebrain neuron-specific L-PGDS knockdown did not alter 90 min MCAO/24 h reperfusion-induced cerebral damage in water-fed mice but significantly attenuated the neuroprotective effect of LAC against cerebral I/R injury (Figure 3C–E).

### 2.4. Effect of Chronic Alcohol Consumption on Neuronal Apoptosis

To examine neuronal apoptosis, double staining of NeuN and TUNEL was performed. TUNEL-positive neuron was not found before ischemic stroke in the brain of either water- or ethanol-fed mice (data not shown). A 90 min MCAO/24 h reperfusion produced TUNEL-positive neurons in all groups (Figure 4 and Figure 5). However, the number of TUNEL-positive neurons in the peri-infarct cortex was significantly less in LAC WT mice compared to water-fed WT mice (Figure 4A,B). Four-week AT-56 administration did not further increase the number of TUNEL-positive neurons in water-fed WT mice but significantly alleviated the inhibitory effect of LAC on post-ischemic neuronal apoptosis (Figure 4A,B). Ninety-minute MCAO/24 h reperfusion-induced neuronal apoptosis in the peri-infarct cortex was also significantly reduced in LAC CaMKIIα^CreERT2/+^/L-PGDS^+/+^ mice compared to water-fed CaMKIIα^CreERT2/+^/L-PGDS^+/+^ mice (Figure 5A,B). The inhibitory effect was abolished by forebrain neuron-specific L-PGDS knockdown (Figure 5A,B). 

### 2.5. Effect of Chronic Alcohol Consumption on Post-Ischemic Cleaved Caspase-3-Positive Neurons

To examine the cleaved caspase-3-positive neurons, double staining of NeuN and cleaved caspase-3 was performed. There was no cleaved caspase-3-positive neuron observed in the brain before ischemic stroke (data not shown). As shown in Figure 6 and Figure 7, a 90 min MCAO/24 h reperfusion produced cleaved caspase-3-positive neurons in all groups. However, the number of cleaved caspase-3-positive neurons in the peri-infarct cortex was significantly reduced in LAC WT mice and CaMKIIα^CreERT2/+^/L-PGDS^+/+^ mice compared to water-fed WT mice and CaMKIIα^CreERT2/+^/L-PGDS^+/+^ mice, respectively. AT-56 administration (Figure 6A,B) and forebrain neuron-specific L-PGDS knockdown (Figure 7A,B) significantly attenuated the inhibitory effect of LAC on post-ischemic cleavage of caspase-3 in cortical neurons of the peri-infarct area.

## 3. Discussion

In the present study, the role of L-PGDS in the neuroprotective effect of LAC against transient focal cerebral ischemia was investigated. There are four new findings. First, LAC upregulated L-PGDS in the cerebral cortex, whereas HAC did not significantly alter the protein expression of L-PGDS in the cerebral cortex. Second, cortical neurons may be also a cellular source of L-PGDS in the forebrain under basal conditions. Third, L-PGDS inhibitor and forebrain neuron-specific L-PGDS knockdown abolished the neuroprotective effect of LAC against cerebral I/R injury. Fourth, L-PGDS inhibitor and forebrain neuron-specific L-PGDS knockdown alleviated the inhibitory effect of LAC on post-ischemic neuronal apoptosis. These findings complement and extend those reported previously [15]. We speculate that LAC may protect against cerebral I/R injury by suppressing post-ischemic apoptosis via the upregulation of L-PGDS.

Light-to-moderate alcohol consumption is defined as drinking up to one and two American standard drinks (14 g of ethanol/each) per day for women and men, respectively. On the other hand, heavy alcohol consumption is specified as drinking more than three and four American standard drinks per day for women and men, respectively. In our previous study, gavage feeding with 0.7 g/kg ethanol raised blood ethanol concentration up to 9 mM, which is commonly observed in a man with average body weight (70 kg) following intake of one and a half American standard drinks. On the other hand, gavage feeding with 2.8 g/kg ethanol increased blood ethanol concentration to 37 mM, which is commonly found in a man with average body weight following consumption of a little more than seven American standard drinks [14]. Thus, gavage feeding with 0.7 and 2.8 g/kg/day ethanol represents LAC and HAC, respectively. The present study is the first to examine the influence of chronic alcohol consumption on L-PGDS in the brain. Very few studies have previously investigated the correlation between alcohol consumption and L-PGDS. Wallenius et al. found that serum level of L-PGDS were inversely correlated with alcohol intake in humans [22]. In contrast, Paulson et al. reported that chronic alcohol consumption increased L-PGDS mRNA in the adipose tissue of mice [21]. In the present study, LAC significantly upregulated L-PGDS, whereas HAC did not alter L-PGDS in the cerebral cortex. Thus, it is possible that L-PGDS may be involved in the protective effect of LAC but not related to the detrimental effect of HAC on ischemic stroke. The precise mechanism by which LAC upregulates L-PGDS in the brain, however, remains to be determined.

L-PGDS is mainly expressed in the CNS and cardiovascular system [16,17]. It is a major protein in human cerebrospinal fluid (CSF). L-PGDS was previously reported to localize in oligodendrocytes, choroid plexus epithelial cells, and leptomeningeal cells under basal conditions and neurons following ischemic stroke [17,20,23]. CaMKII is a unique neuronal signaling protein [24]. CaMKIIα is mainly located in the hippocampus and neocortex of the forebrain [25]. It has been used as a marker gene for excitatory neurons [26]. The CreERT2 fusion gene is tamoxifen inducible. Following tamoxifen administration, Cre recombinase expression is turned on to delete loxP-flanked DNA sequences in target cells. In the present study, tamoxifen administration remarkably reduced L-PGDS protein expression in the cerebral cortex of tamoxifen-treated CaMKIIα^CreERT2/+^/L-PGDS^flox/flox^ mice. It is conceivable that L-PGDS may also localize in the cortical neurons and/or be secreted from the cortical neurons under basal conditions.

The biological function of L-PGDS is far from clear. To date, it has been confirmed to have two functions, catalyzing the conversion of PGH_2_ to PGD_2_ and transporting a variety of small lipophilic molecules, including retinoids and bilirubin [27]. A very few studies investigated the role of L-PGDS in ischemic stroke. Saleem et al. reported that global L-PGDS knockout significantly increased infarct volume and neurological deficits following either transient or permanent ischemic stroke [19]. Gonzalez-Rodriguez et al. reported that L-PGDS was involved in dexamethasone-induced neuroprotection against neonatal hypoxic-ischemic brain injury [28]. Recently, Zhang et al. found that L-PGDS was upregulated in the peri-infarct cortex following transient focal cerebral ischemia in rats. In addition, shRNA-mediated downregulation of L-PGDS significantly exacerbated cerebral I/R injury [20]. Thus, it appears that L-PGDS plays a protective role following ischemic stroke. In the present study, however, although both AT-56 and forebrain neuron-specific L-PGDS knockdown abolished the neuroprotective effect of LAC, neither AT-56 nor forebrain neuron-specific L-PGDS knockdown significantly altered infarct volume and neurological deficits in nonalcohol-fed mice. The reasons for the discrepancies under basal conditions between these studies are not entirely clear but may be related to the animal models and dose and duration of AT-56 treatment utilized in the present study. In addition to neurons, other cellular sources contribute to the L-PGDS content in the brain. Oral administration of AT-56 at a dose of 30 mg/kg has been reported to effectively inhibit L-PGDS activity to 40% in the brain [29]. In addition, chronic treatment with AT-56 may result in a compensatory increase in L-PGDS activity. Thus, either forebrain neuron-specific L-PGDS knockdown or four-week treatment with 30 mg/kg/day AT-56 only partially reduced L-PGDS activity in the brain. If the relationship between L-PGDS expression/activity and its protective effect is a sigmoidal curve, it can be suggested that mild-to-moderate reduction in L-PGDS expression/activity does not significantly affect its protective effect. 

Apoptosis is an essential mechanism involved in cerebral I/R injury. In a previous study, Taniike reported that L-PGDS protects neurons and glial cells against glycosylsphingoid psychosine-induced apoptosis [30]. Recently, Zhang et al. found that vagus nerve stimulation-induced suppression in post-ischemic neuronal apoptosis was related to an increased L-PGDS [20]. In the present study, both AT-56 and forebrain neuron-specific L-PGDS knockdown abolished the anti-apoptotic effect of LAC. These findings suggest that L-PGDS may protect neurons against apoptosis under certain pathological conditions. One of the triggers for post-ischemic apoptosis is reactive oxygen species (ROS) generation due to the sudden availability of oxygen to tissue previously deprived of oxygen and glucose. Mitochondria are the major source of ROS following ischemic stroke. Mitochondrial ROS block mitochondrial respiration and facilitate mitochondrial transition pore formation, which leads to apoptosis via the release of inner and outer mitochondrial membrane components, including cytochrome c and apoptosis-inducing factor (AIF). In cultured human neuroblastoma SH-SY5Y cells, L-PGDS prevents H_2_O_2_-induced neuronal cell death. Furthermore, the protective effect of L-PGDS may be related to its scavenging effect on ROS [31]. L-PGDS catalyzes the conversion of PGH_2_ to PGD_2_. The non-enzymatic metabolites of PGD2, such as 15d-PGJ_2_, 15d-PGD_2_, and Δ^12^-PGJ_2_, are ligands of peroxisome proliferator-activated receptor-gamma (PPARγ) [32]. Recently, we found that LAC increases nuclear PPARγ expression and DNA-binding activity in the brain [33,34]. Moreover, LAC may upregulate manganese superoxide dismutase (MnSOD) via an increased PPARγ activation [34]. MnSOD is the major mitochondria-resident antioxidant enzyme that detoxifies superoxide anions produced by the mitochondrial respiratory chain. Thus, we speculate that the neuroprotective effect of LAC against post-ischemic neuronal apoptosis may be related to the direct and indirect inhibitory effects of L-PGDS on oxidative stress.

In summary, the present study was the first to determine the role of L-PGDS in the neuroprotective effect of LAC against cerebral I/R injury. We found that LAC upregulated L-PGDS in the brain. In addition, L-PGDS inhibitor and forebrain neuron-specific L-PGDS knockdown attenuated the LAC-induced reduction in infarct volume and neuronal apoptosis following transient focal cerebral ischemia. Therefore, chronic ethanol consumption is an important factor implicated in the pathophysiology of transient focal cerebral ischemia. Understanding how alcohol affects an ischemic stroke may not only lead to novel strategies to prevent and treat ischemic stroke in non-alcohol users with a high risk of developing ischemic stroke but will also advance the clinical management of ischemic stroke in alcohol users.

## 4. Materials and Methods

### 4.1. Animal Models

All of the procedures and protocols were approved (September 8, 2015) by the Institutional Animal Care and Use Committee (IACUC) at the Louisiana State University Health Science Center (LSUHSC)-Shreveport and performed following the National Institutes of Health Guide for the Care and Use Laboratory Animals and the ARRIVE (Animal Research: Reporting in Vivo Experiments) guidelines. Male C57BL/6J wild-type (WT) (20–25g, 10–12 weeks), tamoxifen-induced forebrain neuronal L-PGDS knockout (CaMKIIα^CreERT2/+^/L-PGDS^flox/flox^) (20–25g, 10–12 weeks), and littermate control (CaMKIIα^CreERT2/+^/L-PGDS^+/+^) (20–25g, 10–12 weeks) mice on a C57/BL6J background were used. Tamoxifen-induced forebrain neuronal L-PGDS knockout mice were generated by crossbreeding male CaMKIIα^CreERT2/+^ mice (Stock #: 012362, Jackson Labs, Bar Harbor, ME, USA) with female L-PGDS^flox/flox^ mice (generated via Cyagen, Santa Clara, CA, USA). Genotypes were determined by polymerase chain reaction (PCR) analysis.

### 4.2. Ethanol Preconditioning and Treatment

Wild-type (WT) mice (n = 30) were randomly divided into four groups: water (n = 9), 0.7 g/kg/day ethanol (n = 9), 2.8 g/kg/day ethanol (n = 4), water + AT-56 (n = 4), and 0.7 g/kg/day ethanol + AT-56 (n = 4). CaMKIIα^CreERT2/+^/L-PGDS^flox/flox^ mice (n = 14) and CaMKIIα^CreERT2/+^/L-PGDS^+/+^ mice (n = 13) were randomly divided into four groups, CaMKIIα^CreERT2/+^/L-PGDS^flox/flox^ + water (n = 8), CaMKIIα^CreERT2/+^/L-PGDS^flox/flox^ + 0.7 g/kg/day ethanol (n = 6), CaMKIIα^CreERT2/+^/L-PGDS^+/+^ + water (n = 8), and CaMKIIα^CreERT2/+^/L-PGDS^+/+^ + 0.7 g/kg/day ethanol (n = 5). Ethanol groups were gavage fed with 10 mL/kg 7% or 28% ethanol once a day for eight weeks. The water groups were gavage fed with 10 mL/kg water. CaMKIIα^CreERT2/+^/L-PGDS^flox/flox^ mice and CaMKIIα^CreERT2/+^/L-PGDS^+/+^ mice were intraperitoneally injected with tamoxifen (2 mg/day) for 5 consecutive days during the 4th week. From the 5th week, AT-56 (30 mg/kg/day), a selective inhibitor of L-PGDS, was orally given to AT-56-treated groups for four weeks. At the end of 8 weeks of feeding, body weight, blood pressure, and heart rate were measured similarly as described previously [14]. Blood pressure and heart rate were measured using a CODA mouse tail-cuff system (Kent Scientific, Torrington, CT, USA). Before the actual measurement, mice were trained for 3 consecutive days to acclimate to the restraint device and to also have the tail-cuff placed on them. 

### 4.3. Transient Focal Cerebral Ischemia 

Transient focal cerebral ischemia was produced by unilateral MCAO for 90 min as described previously [14]. To avoid the possible effect of acute alcohol, ethanol was not given on the day of MCAO or after MCAO. Prior to the procedure, mice were anesthetized with isoflurane (induction at 5% and maintenance at 1.5%) in a gas mixture containing 30% O_2_/70% N_2_ via a facemask. Body temperature was maintained with a temperature-controlled heating pad (Harvard Apparatus, March, Germany) during the experiment. A laser-Doppler flow probe (PERIMED, PF 5010 LDPM Unit, Jarfalla, Sweden) was attached to the right side of the dorsal surface of the skull to monitor regional cerebral blood flow (rCBF). The right common and external carotid arteries were exposed and ligated. The right middle cerebral artery (MCA) was occluded by inserting a silicon rubber-coated monofilament (Doccol Corporation, Sharon, MA, USA) from the basal part of the right external carotid artery and advanced cranially into the right internal carotid artery to the point where the right middle cerebral artery branched off from the right internal artery. The success of the MCAO was indicated by an immediate drop in rCBF. After the occlusion of the right MCA for 90 min, reperfusion was achieved by withdrawing the suture and reopening the right common carotid artery. Animals were allowed to recover for 24 h. A 24-point scoring system was used to evaluate sensorimotor deficits as described previously [14]. Six tests (spontaneous activity, symmetry of movement, floor walking, beam walking, symmetry of forelimbs, and climbing wall of wire cage) for motor function and two tests (response to vibrissae touch and reaction to touch on either side of trunk) for sensory function were graded on a scale of 0 to 3 each. Neurological scores were assigned as follows: 0, complete deficit; 1, the definite deficit with some function; 2, decreased response or mild deficit; 3, no evidence of deficit/symmetrical responses [35].

### 4.4. Brain Collection and Processing

The brains were collected and processed as described previously [14]. Eighteen mice (water-fed WT mice (n = 5), 0.7 g/kg/d ethanol-fed WT mice (n = 5), 2.8 g/kg/d ethanol-fed WT mice (n = 4), tamoxifen-treated CaMKIIα^CreERT2/+^/L-PGDS^flox/flox^ mice (n = 2), and tamoxifen-treated CaMKIIα^CreERT2/+^/L-PGDS^+/+^ mice (n = 2)) were euthanized and exsanguinated. The brains were removed and cut into six 1.75 mm-thick coronal sections. Cortical tissues were collected from the parietal cortex. The remaining mice were anesthetized and perfused transcardially with 1X phosphate-buffered saline (PBS), followed by 4% paraformaldehyde in 0.1 M PBS. The brains were removed, fixed overnight in 4% paraformaldehyde in 0.1 M PBS, dehydrated in a graded series of sugar solutions for 72 h, then embedded in O.C.T. compound (Fisher Scientific, Hampton, NH, USA), and quickly frozen for 5 min in liquid nitrogen. The frozen brains were then cut into 14 μm coronal sections and placed on frost-free slides. 

### 4.5. Nissl Staining

To measure the infarct size, sections from eight levels (between 2.9 mm rostral and 4.9 mm caudal to bregma) were selected for Nissl staining [36]. The sections were incubated in 0.01% cresyl violet acetate (Sigma, St. Louis, MO, USA) solution for 14 min at 60 °C, rinsed in distilled water, dehydrated in ethanol, cleaned in xylene, and covered with xylene-based mounting media (VWR). The sections were photographed under 1.0× magnification (Olympus, Tokyo, Japan) and evaluated using ImageJ. A complete lack of staining defined the infarct lesion. The infarct size was calculated by integration of the area of infarct lesion with the distance between coronal levels and expressed as a percentage of total hemispheric volume. 

### 4.6. Neuronal Terminal Deoxynucleotidyl Transferase dUTP Nick end Labeling (TUNEL) Staining

Three sections (1.21 mm rostral and 0.23 mm and 1.31 mm caudal to bregma) from each mouse were used to evaluate the neural apoptosis. The sections were washed with 1X PBS, blocked with 10% bovine serum albumin (BSA) for at least 1 h, and then incubated overnight at 4 °C with 1:100 rabbit anti-NeuN (Millipore, Burlington, MA, USA) for visualization of the neuron as primary antibodies. The sections were then incubated with 1:200 AlexaFluor 546 donkey anti-rabbit (Invitrogen, Waltham, MA, USA) for one hour at room temperature. In situ cell death detection kit (Roche, Basel, Switzerland) was used according to the manufacturer’s instruction for TUNEL staining. Briefly, the brain sections were incubated in permeabilization solution containing 0.1% Triton X-100 and 0.1% sodium citrate for 20 min at room temperature. Then, the sections were incubated with a TUNEL reaction mixture for 1 h at 37 °C, mounted with DAPI mounting medium (Vector, Burlingame, CA, USA), and observed under a fluorescence microscope (Olympus). The area with dramatically reduced NeuN staining was defined as the infarct core. For quantitative analysis, both TUNEL and NeuN positive cells were counted in one parietal cortex area and two temporal cortex areas surrounding the infarct core per section and expressed as percentage change to the control group.

### 4.7. Double Immunohistochemical Staining

Three sections (1.21 mm rostral and 0.23 mm and 1.31 mm caudal to bregma) from each mouse were washed with 1X PBS, blocked with 10% bovine serum albumin (BSA) for at least 1 h, and then incubated overnight at 4 °C with mouse anti-NeuN (Millipore, Burlington, MA, USA) and 1:100 rabbit anti-cleaved caspase-3 (Cell Signaling, Danvers, MA, USA). The sections were then incubated with 1:200 AlexaFluor 546 goat anti-mouse and AlexaFluor 488 goat anti-rabbit (Invitrogen, Waltham, MA, USA) for one hour at room temperature. Finally, sections were mounted with DAPI mounting medium with a Vector shield and visualized using a Nikon fluorescence microscope (Eclipse Ts2) (Melville, NY, USA). For quantitative analysis, both NeuN and cleaved caspase-3 positive cells were counted in one parietal cortex area and two temporal cortex areas surrounding the infarct core per section and expressed as percentage change to the control group.

### 4.8. Western Blot Analysis

Western blot experiments were conducted as described previously [14]. SDS polyacrylamide gel electrophoresis (SDS-PAGE) was performed on a 10% gel on which 20–30 μg of total protein per well was loaded. After SDS-PAGE, the proteins were transferred onto a polyvinylidene difluoride membrane. Immunoblotting was performed with rabbit anti-L-PGDS (Novus, Abingdon, UK) and mouse anti-GAPDH as the primary and peroxidase-conjugated goat anti-mouse and goat anti-rabbit IgG as the second antibody. The bound antibody was detected by enhanced chemiluminescence (ECL) detection (Pierce Chemical and Genesee, Dallas, TX, USA), and the bands were analyzed using ChemiDocTM MP Imaging System (Bio-Rad, Hercules, CA, USA). For quantification, protein expression was normalized with the loading control and expressed as percentage changes to the control group.

### 4.9. Statistical Analysis

Data are reported as means ± SEM. The difference in baseline L-PGDS was analyzed with one-way ANOVA with Dunnett’s post hoc test. The differences in infarct size, neurological score, TUNEL-positive neuron, and cleaved caspase-3-positive neuron between groups were analyzed with two-way ANOVA followed by Tukey’s test for significance. A *p* value of 0.05 or less was considered to be significant.

## Figures and Tables

**Figure 1 ijms-23-00133-f001:**
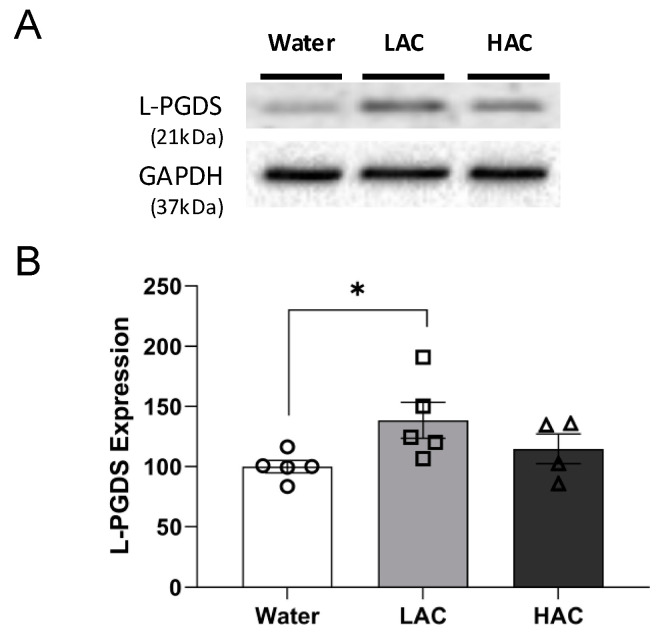
Effect of chronic ethanol consumption on protein expression of L-PGDS in the cerebral cortex. (**A**) Representative Western blots. (**B**) Values are means ± SEM for 4–5 mice in each group. * *p* < 0.05. Analyzed using one-way ANOVA with Dunnett’s post hoc test.

**Figure 2 ijms-23-00133-f002:**
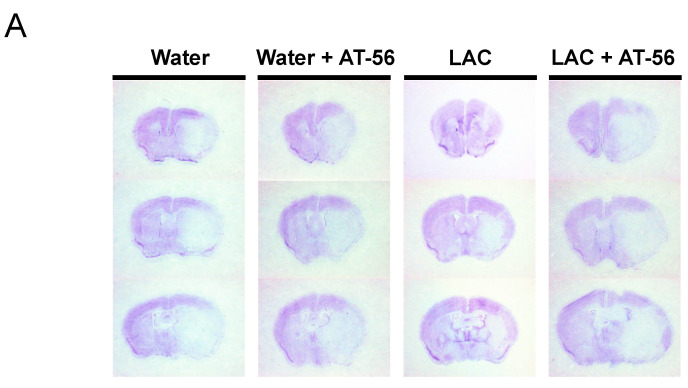
Effect of AT-56 on cerebral I/R injury. (**A**) Representative brain sections stained with cresyl violet. (**B**) Total infarct volume. (**C**) Neurological deficit score. Values are means ± SEM for 4 mice in each group. * *p* < 0.05. ** *p* < 0.005. Analyzed using two-way ANOVA followed by Tukey’s test.

**Figure 3 ijms-23-00133-f003:**
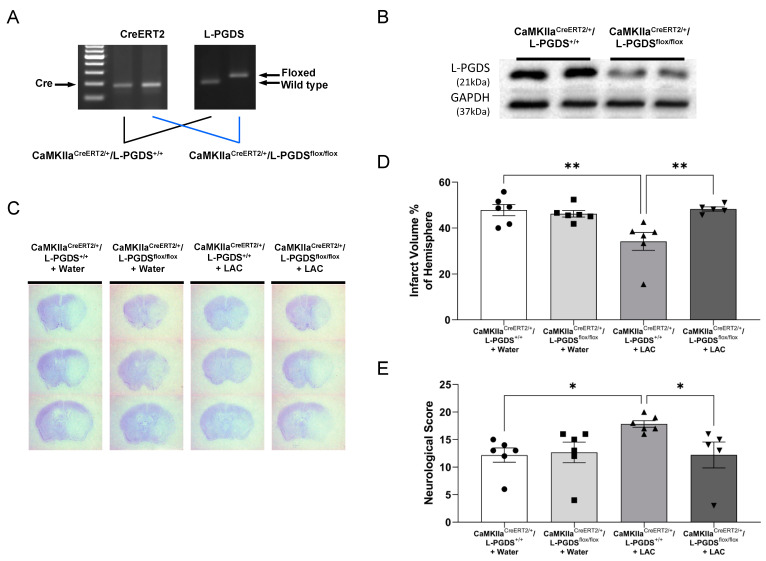
Effect of forebrain neuron-specific L-PGDS knockdown on cerebral I/R injury. (**A**) PCR-based genotyping. The homozygous (fl/fl) mice with Cre positive were considered to be forebrain neuron-specific L-PGDS knockdown mice following tamoxifen treatment. (**B**) Western blots of L-PGDS in the cerebral cortex of tamoxifen-treated CaMKIIα^CreERT2/+^/L-PGDS^+/+^ and CaMKIIα^CreERT2/+^/L-PGDS^flox/flox^ mice. (**C**) Representative brain sections stained with cresyl violet. (**D**) Total infarct volume. (**E**) Neurological deficit score. Values are means ± SEM for 5–6 mice in each group. * *p* < 0.05. ** *p* < 0.005. Analyzed using two-way ANOVA followed by Tukey’s test.

**Figure 4 ijms-23-00133-f004:**
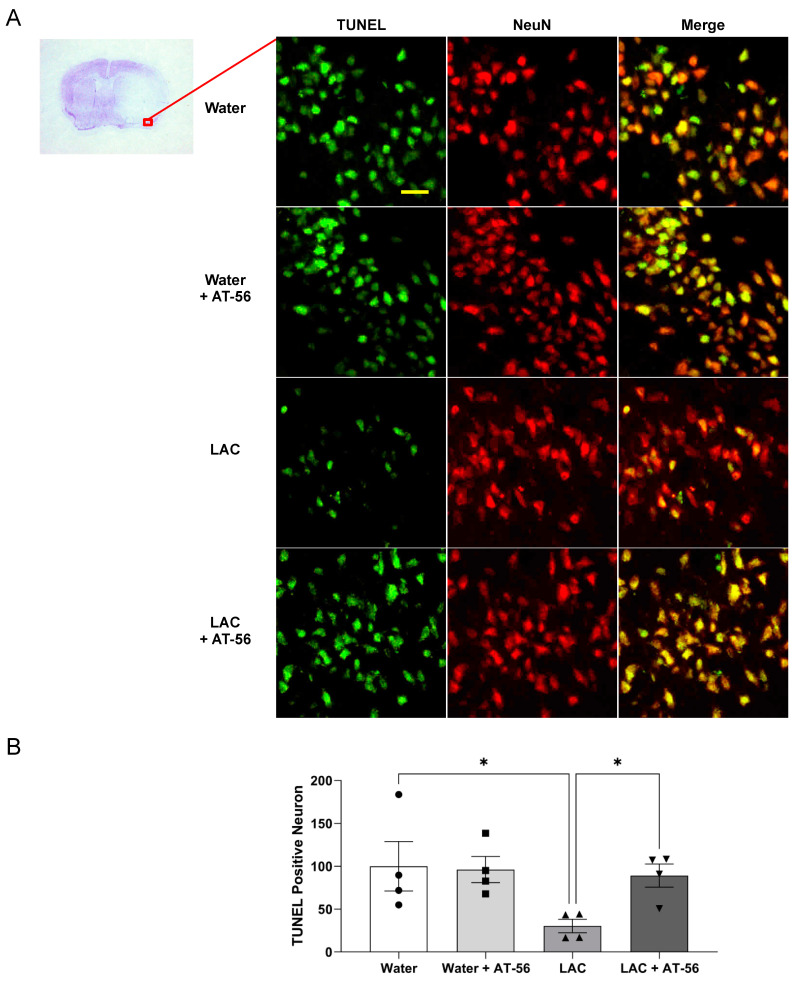
Effect of AT-56 on TUNEL-positive neurons in the cerebral cortex following transient focal cerebral ischemia. (**A**) Representative double staining of NeuN and TUNEL in the peri-infarct cortex of the temporal lobe at the section 0.23 mm caudal to bregma. Scale bar = 20 μm. (**B**) Values of TUNEL-positive neurons are means ± SEM for 4 mice in each group. * *p* < 0.05. Analyzed using two-way ANOVA followed by Tukey’s test.

**Figure 5 ijms-23-00133-f005:**
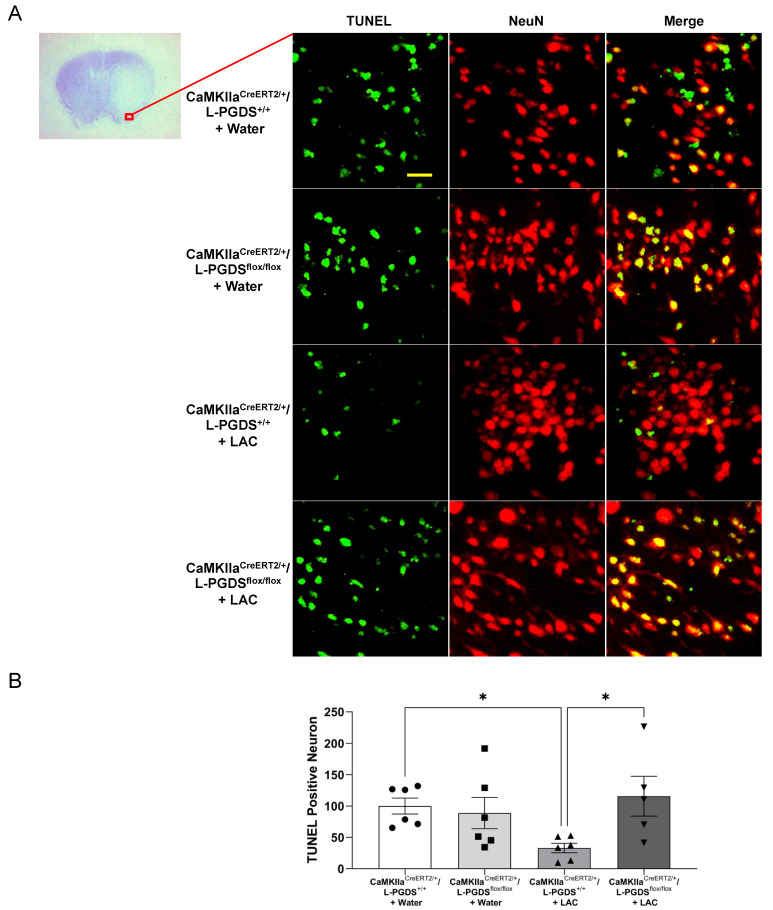
Effect of forebrain neuron-specific L-PGDS knockdown on TUNEL-positive neurons in the cerebral cortex following transient focal cerebral ischemia. (**A**) Representative double staining of NeuN and TUNEL in the peri-infarct cortex of the temporal lobe at the section 0.23 mm caudal to bregma. Scale bar = 20 μm. (**B**) Values of TUNEL-positive neurons are means ± SEM for 5–6 mice in each group. * *p* < 0.05. Analyzed using two-way ANOVA followed by Tukey’s test.

**Figure 6 ijms-23-00133-f006:**
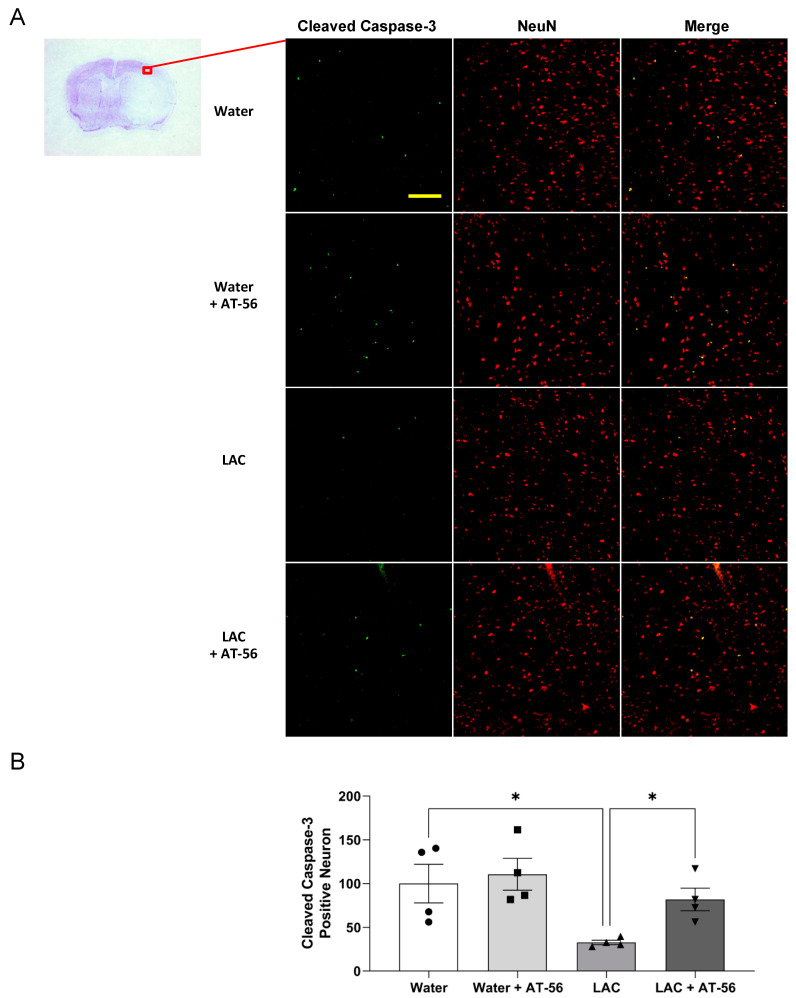
Effect of AT-56 on cleaved caspase-3-positive neurons in the cerebral cortex following transient focal cerebral ischemia. (**A**) Representative double staining of NeuN and cleaved caspase-3 in the peri-infarct cortex of the parietal lobe at the section 0.23 mm caudal to bregma. Scale bar = 100 μm. (**B**) Values of cleaved caspase-3-positive neurons are means ± SEM for 4 mice in each group. * *p* < 0.05. Analyzed using two-way ANOVA followed by Tukey’s test.

**Figure 7 ijms-23-00133-f007:**
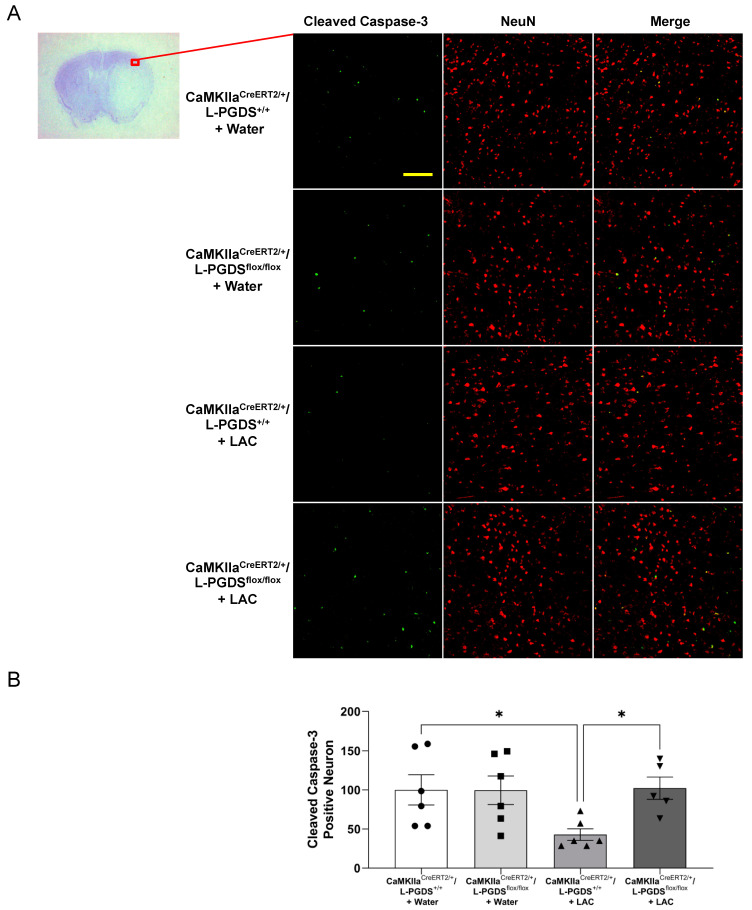
Effect of forebrain neuron-specific L-PGDS knockdown on the cleaved caspase-3-positive neuron in the cerebral cortex following transient focal cerebral ischemia. (**A**) Representative double staining of NeuN and TUNEL in the peri-infarct cortex of the parietal lobe at the section 0.23 mm caudal to bregma. Scale bar = 100 μm. (**B**) Values of cleaved caspase-3-positive neurons are means ± SEM for 5–6 mice in each group. * *p* < 0.05. Analyzed using two-way ANOVA followed by Tukey’s test.

**Table 1 ijms-23-00133-t001:** Physiological parameters.

	Body Weight (g)	MABP (mmHg)	Heart Rate (bpm)
Water	27 ± 1	82 ± 5	637 ± 40
Water + AT-56	28 ± 1	88 ± 1	657 ± 77
LAC	27 ± 1	82 ± 3	639 ± 31
LAC + AT-56	28 ± 1	88 ± 2	674 ± 13
HAC	28 ± 1	86 ± 2	654 ± 18
CaMKIIa^CreERT2/+^/L-PGDS^+/+^ + Water	26 ± 2	90 ± 4	647 ± 4
CaMKIIa^CreERT2/+^/L-PGDS^flox/flox^ + WaterCaMKIIa^CreERT2/+^/L-PGDS^+/+^ + LAC	27 ± 1	87 ± 6	691 ± 27
	26 ± 1	89 ± 3	643 ± 11
CaMKIIa^CreERT2/+^/L-PGDS^flox/flox^ + LAC	27 ± 2	89 ± 2	694 ± 21

## Data Availability

The data presented in this study are available on request from the corresponding author.

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
