# Peer review of "Protective Effect of Low-Dose Alcohol Consumption against Post-Ischemic Neuronal Apoptosis: Role of L-PGDS"

_ijms, 2021, doi:10.3390/ijms23010133_

Round 1

Reviewer 1 Report

In this paper, the authors tried to evidence that the protection of neuronal cells against ischemia/reperfusion injury by administration of low ethanol amounts is mediated by lipocalin type prostaglandin D2 synthase. The results of experiments using conditional/neuronal specific KO mice and the lipocalin-type PGD2 synthase inhibitor AT-56 are quite convincing. However, in these experiments, it is noticeable that there is no effect of PGD2 synthase inhibition or KO in absence of ethanol administration. The authors discussed this point in lines 161-165 and postulated that AT-65 only partially inhibits PGD2 activity (about 40% inhibition) and this partial inhibition could explain this absence of effect in apoptosis level in control condition. Since a 50% reduction of PGD2 expression by KO is shown in figure 3B, I suppose that the hypothesis is the same for the effect in KO mice but the authors should mention it. However the authors show in figure 1 that gavage of mice with low amounts of ethanol increase PGD2 synthase expression by 40%. Therefore, it is hard to understand why a 40% increase of expression levels lead to clear effects in the case of ethanol administration while a 40% reduction has no effect in control conditions. The author should provide other hypothesis to explain their results. The effects of ethanol administration on PGD2 expression levels are measured in western blotting experiments. Did the authors perform immunochemistry experiments to investigate whether the ethanol effect is homogenous or restricted to some neuronal cells? Furthermore, since AT-65 is known to bind to the catalytic site of PGD2 synthase, the variations of PGD2 synthase expression levels should also result in modifications of PGD2 production. However, the authors did not measure PGD2 concentration in brain or CSF. PGD2 measurements could be useful to confirm that enzymatic activity of PGD2 synthase is involved in the reduction of apoptosis levels by ethanol consumption. They evoke on lines 153-154 that PGD2 synthase is also a transporter for bilirubin and lipophilic molecules. The authors should clarify their hypothesis and explain how they reconciliate the transport function of PGD2 synthase with the observed effects of AT-65. Finally, the authors showed that high amounts of ethanol did not increase lipocalin type PGD2 synthase expression in figure 1. But they did not display any result about the effect of gavages with high ethanol amounts in later experiments. It is important to clarify the context in which the authors situate their work. Do they want to indicate that excessive alcohol consumption cancels out protective effects? Or do they wish to recommend the administration of low doses of ethanol in strokes? This should be clearly stated in the introduction. In the first case, the authors should add additional experiment on the effects of high ethanol amounts.

I have also some minor comments. The font size does not appear uniform in the text. For example, KO animals are indicated with a larger font and the reference 36 is written with a smaller font. In legends SE means SD or SEM?

Author Response

  1. In this paper, the authors tried to evidence that the protection of neuronal cells against ischemia/reperfusion injury by administration of low ethanol amounts is mediated by lipocalin type prostaglandin D2 synthase. The results of experiments using conditional/neuronal specific KO mice and the lipocalin-type PGD2 synthase inhibitor AT-56 are quite convincing. However, in these experiments, it is noticeable that there is no effect of PGD2 synthase inhibition or KO in absence of ethanol administration. The authors discussed this point in lines 161-165 and postulated that AT-65 only partially inhibits PGD2 activity (about 40% inhibition) and this partial inhibition could explain this absence of effect in apoptosis level in control condition. Since a 50% reduction of PGD2 expression by KO is shown in figure 3B, I suppose that the hypothesis is the same for the effect in KO mice but the authors should mention it. However the authors show in figure 1 that gavage of mice with low amounts of ethanol increase PGD2 synthase expression by 40%. Therefore, it is hard to understand why a 40% increase of expression levels lead to clear effects in the case of ethanol administration while a 40% reduction has no effect in control conditions. The author should provide other hypothesis to explain their results.

We have added a discussion to address why AT-56 and forebrain neuron-specific L-PGDS knockdown did not affect the infarct volume and neuronal apoptosis in water-fed mice. Please see page 11.

  1. The effects of ethanol administration on PGD2 expression levels are measured in western blotting experiments. Did the authors perform immunochemistry experiments to investigate whether the ethanol effect is homogenous or restricted to some neuronal cells? Furthermore, since AT-65 is known to bind to the catalytic site of PGD2 synthase, the variations of PGD2 synthase expression levels should also result in modifications of PGD2 production. However, the authors did not measure PGD2 concentration in brain or CSF. PGD2 measurements could be useful to confirm that enzymatic activity of PGD2 synthase is involved in the reduction of apoptosis levels by ethanol consumption. They evoke on lines 153-154 that PGD2 synthase is also a transporter for bilirubin and lipophilic molecules. The authors should clarify their hypothesis and explain how they reconciliate the transport function of PGD2 synthase with the observed effects of AT-65.

We did perform immunostaining of L-PGDS. We tried many L-PGDS primary antibodies and found that only choroid plexus epithelial cells are L-PGDS-positive in the brain under basal conditions as well as following ischemic stroke. Since we found a diffuse background staining, we generated CaMKIIαCreERT2/+/L-PGDSflox/flox mice. Upon tamoxifen treatment,  L-PGDS expression was largely reduced in CaMKIIαCreERT2/+/L-PGDSflox/flox mice, indicating that neurons may be a major source of L-PGDS in the brain. As described in the discussion, L-PGDS can protect against neuronal apoptosis via either its enzymatic or nonenzymatic effect. L-PGDS requires free sulfhydryl compounds, such as β-mercaptoethanol, DTT, or GSH, for its catalytic reaction. Thus, an upregulated L-PGDS does not mean an increased L-PGDS activity. In addition, PGD2 is chemically unstable and nonenzymatically dehydrated to produce the J series of PGs with a cyclopentenone structure. Therefore, it is likely that PGD2 does not increase in the brain of alcohol-fed mice. Moreover, the present study focused on L-PGDS. We believe that studying PGD2 is beyond the scope of the present study and thus did not measure PGD2.

  1. Finally, the authors showed that high amounts of ethanol did not increase lipocalin type PGD2 synthase expression in figure 1. But they did not display any result about the effect of gavages with high ethanol amounts in later experiments. It is important to clarify the context in which the authors situate their work. Do they want to indicate that excessive alcohol consumption cancels out protective effects? Or do they wish to recommend the administration of low doses of ethanol in strokes? This should be clearly stated in the introduction. In the first case, the authors should add additional experiment on the effects of high ethanol amounts.

In our previous studies, we have shown that excessive alcohol consumption exacerbates ischemic brain damage, which is also consistent with the results of epidemiological studies. Since excessive alcohol consumption did not alter L-PGDS in the brain, we speculate that L-PGDS may not play a role in the detrimental effect of excessive alcohol consumption. Whereafter, we focused on the role of L-PGDS in the protective effect of LAC. We have added the reason in the introduction. Please see page 5. It has to be noted that we don’t recommend drinking low-dose alcohol under any circumstances. As indicated in the summary of page 13, we would like to state that chronic ethanol consumption is an important factor implicated in the pathophysiology of transient focal cerebral ischemia. Understanding how alcohol affects an ischemic stroke may not only lead to novel strategies to prevent and treat ischemic stroke in non-alcohol users with a high risk of developing ischemic stroke but also will advance clinical management of ischemic stroke in alcohol users.

  1. I have also some minor comments. The font size does not appear uniform in the text. For example, KO animals are indicated with a larger font and the reference 36 is written with a smaller font. In legends SE means SD or SEM?

The font has been corrected. We have changed SE to SEM.

We would like to thank this reviewer for his/her helpful comments.

Reviewer 2 Report

Dear Authors,

Overview and general recommendation:
I read your research manuscript with great interest. I think it is important to clarify not only the effects of low volume alcohol consumption on ischemic brain disease, which accounts for the majority of cerebrovascular disorder, but also to clarify that pathophysiology because there is still no consensus on best treatment with the ischemic brain injury showing high sequelae rate even developed endovascular surgery and iv-tPA today.

In recent years, some famous journals such as the Lancet and others have reported that chronic consumption of alcohol, even at low doses, is clinically harmful to the body as a whole, and that it promotes apoptosis in liver metabolism. Therefore, the image that alcohol consumption is bad for health even at low doses is gaining ground worldwide. On the other hand, as the authors state in the text, there have been reports in basic and clinical research that low amounts of alcohol are useful in the prevention and treatment of some diseases. And ischemic stroke is one of them. In light of the above, we believe that this study, as well as the previous studies by the authors, can be one of the most socially significant studies.

However, I recommend you had better add some experiments to the research, which will reinforce your point. And there are some things that I think we have to clarify about the research:

Major comments:

  1. The hypotheses of this study are (1) chronic alcohol intake alters the expression of L-PGDS in the brain, and (2) L-PGDS is involved in the inhibition of neuronal apoptosis after transient focal cerebral ischemia induced by LAC. In total, I agree with the authors' results and their interpretation. The authors' setting of LAC and HAC volumes is also based on their previous studies, and I agree with their methodology. In this study, the authors set 24 hours as the outcome time point and examined brain injury especially apoptosis from some molecules. While I agree with the authors' decision to use 24 hours as the outcome point (all three of their previous publications use 24 hours as the outcome; References 13-15.), I do not agree with emphasizing the results at this point in time and with this number of mice (as I will discuss in the next section).

It is common to measure the number of neurons and the infarcted area when examining the effect of treatment on brain protection, as in this study. However, within a few days, neurons may appear to be viable, but may later die and become infarcted. The reasons for this include delayed infarction after mild ischemia, delayed infarct expansion, and delayed cell death due to treatment (temporary protection). Therefore, it is expected that the true infarct size and apoptotic area may vary depending on the time of verification. In this study, the change in infarct size and the change in apoptosis around the infarct are the most important points.From the viewpoint of animal care, it is essential to minimize the sacrifice of mice, but it would be a mistake if minimizing the sacrifice of mice leads to a misleading answer and causes more confusion in interpretation. In order to clarify the effect of low volume alcohol consumption on infarction and its relation to L-PGDS, I think it would be better to show the outcome at another time point (2-time point). The recommended time would be a few days to a week after the results for the reasons mentioned above.

However, it would be difficult to add another time point experiment in all the figures. Therefore, I would recommend adding Fig.1B and Fig.2B. The reason for this is that the three previous publications by the authors have not confirmed the relationship between the infarction lesion and the LAC at any time point other than 24h, to clarify one of the most important aspects of this study, the relationship between LAC and L-PGDS in stroke/reperfusion injury, and because of the nature of the study design of this study.

  1. In order to examine the hypothesis, the authors divided the steps into two groups and examined each group. I agree with most of these methods. However, I think it is necessary to reconsider whether the number of animals for each validation is appropriate, based on the 3R's principle supported by the USDA.In general, the number of animals required for outcome validation is not specified. In my opinion, many researchers use n=5-10 for such experiments and submit a manuscript to the reviewer.

One of the assumptions in conducting a test is that the data are normally distributed. For a very small sample, this is probably not true. There is no clear definition of "very small sample", but looking at the graph, we can see that it is difficult to analyze accurately with a small sample. For example, in Figure 1B, despite the small number of n=4-5, at first glance it seems that the LAC group is affected by the large L-PGDS expression of one mouse. Furthermore, in Figure.2B, the LAC group is split into two small infarct and two medium infarct groups, and it seems that if we add more animals, we can derive more reliable results by converging to one of them. As far as the number of animals is concerned, the above seems to apply to some extent to the other figures as well (all three of the authors' previous reports had n=5 and more).

In summary, although Figures 1and2 are complementary work to previous reports, they are important experiments that provide the basis for this study, so we believe that the addition of the number of animals used in Figures 1B and 2B is necessary (acceptable for n=5 or more). In addition, we recommend that the number of animals used in each group, including the other figures, should be aligned and additional animals should be considered if possible.

Minor comments:

  1. 2, comparing B and C, the colors of the bar graphs of each group do not seem to correspond to each other. If they do not correspond, proceed with the correction (LAC+AT group and others, if any.)
  2. 3 D and E also look the same as above, but the colors of the bar graphs for each group do not seem to correspond. (Also CaMKIICreERKT2/+/L-PGDSflox/flox+LAC group and CaMKIICreERKT2/+/L-PGDSflox/flox+Water group? )

Author Response

  1. It is common to measure the number of neurons and the infarcted area when examining the effect of treatment on brain protection, as in this study. However, within a few days, neurons may appear to be viable, but may later die and become infarcted. The reasons for this include delayed infarction after mild ischemia, delayed infarct expansion, and delayed cell death due to treatment (temporary protection). Therefore, it is expected that the true infarct size and apoptotic area may vary depending on the time of verification. In this study, the change in infarct size and the change in apoptosis around the infarct are the most important points. From the viewpoint of animal care, it is essential to minimize the sacrifice of mice, but it would be a mistake if minimizing the sacrifice of mice leads to a misleading answer and causes more confusion in interpretation. In order to clarify the effect of low volume alcohol consumption on infarction and its relation to L-PGDS, I think it would be better to show the outcome at another time point (2-time point). The recommended time would be a few days to a week after the results for the reasons mentioned above. However, it would be difficult to add another time point experiment in all the figures. Therefore, I would recommend adding Fig.1B and Fig.2B. The reason for this is that the three previous publications by the authors have not confirmed the relationship between the infarction lesion and the LAC at any time point other than 24h, to clarify one of the most important aspects of this study, the relationship between LAC and L-PGDS in stroke/reperfusion injury, and because of the nature of the study design of this study.

This reviewer suggests conducting an additional experiment to measure the infarct lesion at a later time point. However, the time point and specific reasons are not clear. In MCAO-induced transient focal cerebral ischemia mice, the infarct volume reaches the maximal at 24 hours of reperfusion. In addition, we chose a 90-minute, which is the longest in the mouse model of ischemic stroke, as the duration of ischemia to ensure severe ischemia.  Furthermore, our findings that LAC significantly reduced and high-dose alcohol consumption (HAC) significantly increased the infarct volume at 24 hours of reperfusion in our previous and present studies are consistent with the results of epidemiological studies. Thus, we always selected 24 hours of reperfusion as the time point to evaluate cerebral I/R damage. We request this reviewer to reconsider whether the additional experiment is essential.  

  1. In order to examine the hypothesis, the authors divided the steps into two groups and examined each group. I agree with most of these methods. However, I think it is necessary to reconsider whether the number of animals for each validation is appropriate, based on the 3R's principle supported by the USDA. In general, the number of animals required for outcome validation is not specified. In my opinion, many researchers use n=5-10 for such experiments and submit a manuscript to the reviewer. One of the assumptions in conducting a test is that the data are normally distributed. For a very small sample, this is probably not true. There is no clear definition of "very small sample", but looking at the graph, we can see that it is difficult to analyze accurately with a small sample. For example, in Figure 1B, despite the small number of n=4-5, at first glance it seems that the LAC group is affected by the large L-PGDS expression of one mouse. Furthermore, in Figure.2B, the LAC group is split into two small infarct and two medium infarct groups, and it seems that if we add more animals, we can derive more reliable results by converging to one of them. As far as the number of animals is concerned, the above seems to apply to some extent to the other figures as well (all three of the authors' previous reports had n=5 and more).

For statistical analysis, we usually design five animals for each group. Since AT-56 is extremely expensive, we performed two n=2 experiments in the present study. In the first n=2 experiment, each group only contained two animals. Since we found the tendency from the first n=2 experiment, we performed the second n=2 experiment, in which each group contained two animals again. Since the results from two n=2 experiments reached statistical significance, we stopped conducting the further experiment. This is the story behind why the number in each group was four in the study of pharmacological approach. Overall, the number is determined by the statistical analysis. In the present study, the neuroprotective effect of LAC was not only shown in LAC wild-type mice but also found in LAC CaMKIIαCreERT2/+/L-PGDS+/+ mice. In addition to the pharmacological approach, the present study also included a genetic approach. Both approaches consistently demonstrated the involvement of L-PGDS. Therefore, we request this reviewer to reconsider whether adding additional animals is essential.     

  1. Figure 2, comparing B and C, the colors of the bar graphs of each group do not seem to correspond to each other. If they do not correspond, proceed with the correction (LAC+AT group and others, if any.) Figures 3 D and E also look the same as above, but the colors of the bar graphs for each group do not seem to correspond. (Also CaMKIICreERKT2/+/L-PGDSflox/flox+LAC group and CaMKIICreERKT2/+/L-PGDSflox/flox+Water group? )

We have adjusted the color of the bar.

We would like to thank this reviewer for his/her helpful comments.

Round 2

Reviewer 1 Report

I thank the authors for their explanations, especially those regarding immunostaining experiments. I recognize that measurement of PGD2 can be a tricky work because of the lack of specificity of immunoassays, the PGD2 instability and the small quantities of mice tissues which can be used. The authors may indeed consider this work beyond the objective of the article, as described in the introduction, even if this is a limitation of this work. The modifications made in the introduction and discussion sections fully answer to my remarks and critics.

Reviewer 2 Report

Title: Protective Effect of Low-Dose Alcohol Consumption Against Post-Ischemic Neuronal Apoptosis: Role of L- 1 PGDS

Dear Authors,

I checked the authors' response and the revised version of the manuscript.
All points that needed to be corrected had been revised.
The authors also stated in a clear manner that additional experiments are unnecessary or difficult for some reasons and asked the reviewers to reconsider the necessity of additional experiments.

In response to the above, I totally understand their opinion.
Therefore, I reconsidered to exempt the additional experiment.

However, I would like to emphasize again that we need to be careful in interpreting the results of only one time point in the transition of various responses after cerebral infarction, and that we also need to be careful in interpreting the results of data that is statistically significant but the number of experimental subjects is very small and seems to be strongly influenced by one animal.

In any case, in view of the situation of the authors, we have decided that it is appropriate to publish the results of this study in the journal.